# Melatonin as a Possible Natural Anti-Viral Compound in Plant Biocontrol

**DOI:** 10.3390/plants12040781

**Published:** 2023-02-09

**Authors:** Josefa Hernández-Ruiz, Manuela Giraldo-Acosta, Amina El Mihyaoui, Antonio Cano, Marino B. Arnao

**Affiliations:** Phytohormones & Plant Development Laboratory, Department of Plant Biology (Plant Physiology), Faculty of Biology, University of Murcia, 30100 Murcia, Spain

**Keywords:** melatonin, resistance disease, virus, plant biocontrol, plant pathogen, biotic stress, plant immunity

## Abstract

Melatonin is a multifunctional and ubiquitous molecule. In animals, melatonin is a hormone that is involved in a wide range of physiological activities and is also an excellent antioxidant. In plants, it has been considered a master regulator of multiple physiological processes as well as of hormonal homeostasis. Likewise, it is known for its role as a protective biomolecule and activator of tolerance and resistance against biotic and abiotic stress in plants. Since infections by pathogens such as bacteria, fungi and viruses in crops result in large economic losses, interest has been aroused in determining whether melatonin plays a relevant role in plant defense systems against pathogens in general, and against viruses in particular. Currently, several strategies have been applied to combat infection by pathogens, one of them is the use of eco-friendly chemical compounds that induce systemic resistance. Few studies have addressed the use of melatonin as a biocontrol agent for plant diseases caused by viruses. Exogenous melatonin treatments have been used to reduce the incidence of several virus diseases, reducing symptoms, virus titer, and even eradicating the proliferation of viruses such as *Tobacco Mosaic Virus*, *Apple Stem Grooving Virus*, *Rice Stripe Virus* and *Alfalfa Mosaic Virus* in tomato, apple, rice and eggplant, respectively. The possibilities of using melatonin as a possible natural virus biocontrol agent are discussed.

## 1. Introduction

Melatonin (*N*-acetyl-5-methoxytryptamine) is an indoleamine derived from tryptophan and produced by several organisms such as animals [1,2,3,4,5,6,7], bacteria [8,9,10,11], fungi [7,12] and plants [7,10,13]. In mammals, it was discovered in 1958 in the pineal gland where it is synthesized [14]. Since then, its chemical structure has been described [15]. Later, in 1995, melatonin was identified in several plants and so-called phytomelatonin [16,17,18].

In animals, melatonin is involved in the regulation of circadian rhythms, mood, locomotor activity, retina physiology and the seasonal behavior, intervening in processes such as reproductive behavior, sleep, food intake, etc. [19,20,21,22,23]. Melatonin regulates circadian rhythms by acting as a light/darkness signal and sending information to the central nervous system to synchronize physiological processes [24,25]. Moreover, at the cellular level, melatonin acts in many cell metabolic processes, from the scavenging of reactive species of oxygen and nitrogen (ROS and RNS) and activating antioxidative enzymes to being a great antioxidant molecule [26,27,28]. Additionally, melatonin seems to intervene in processes related to the immune system since it can be influenced by light signals through the neuroendocrine. In this way, melatonin acts as an anti-inflammatory, antioxidant and neuroprotective agent in many diseases such as neurodegenerative (dementia, Alzheimer, Parkinson), cardiovascular, obesity, cancer and other dysfunctions [29,30,31,32,33,34,35,36,37,38]. In addition, melatonin has been used to combat animal and human viral diseases, such as Venezuelan equine encephalomyelitis (VEE) virus [39] and Ebola virus diseases [40]. On the other hand and as a consequence of the global pandemic caused by SARS-CoV-2, melatonin has been used in various studies as an active or adjuvant drug for use at different stages of therapy for COVID-19 [41]. Melatonin is not virucidal but it has indirect antiviral actions due to its anti-inflammatory, antioxidant and immune enhancing properties [42].

Concerning plants, many advances to understand the role of phytomelatonin have been made since its discovery in 1995. Its wide ranging biological actions (pleiotropic compound) in plants has led to it being called a multi-regulatory molecule or plant master regulator [43,44]. These actions include the ability to regulate plant growth, rooting, leaf senescence, photosynthetic efficiency and biomass yield, as well as a role as a regulator of flowering, parthenocarpy and fruit and seed ripening [45,46,47,48,49,50,51,52,53]. One of the most studied aspects related to phytomelatonin has been its role as a protective agent against biotic and abiotic stress situations in plants [54,55,56,57,58,59,60,61,62,63,64,65,66]. Because infections by pathogens such as bacteria, fungi and viruses in crops lead to large economic losses, interest arose in knowing if melatonin plays a relevant role in defense systems against pathogens in plants [67,68]. Currently, several strategies have been applied to combat pathogen infections in plants and reduce economic losses. Some examples of these strategies are genetic transformation [69], the use of virus-free plant materials [70] and chemical priming to induce resistance/tolerance of some cultivars to viruses, among others [71].

In this review, we present the main functions of phytomelatonin in plants and its ability to be used as a natural compound against plant pathogen infections, specially focused on the plant antiviral responses. Furthermore, we propose a general mechanism of action to phytomelatonin as a plant antiviral agent and the possibility of its use in agricultural and biotechnological practices, not only for its protective role against virus infections, but for its status as a natural and non-toxic (ecological) molecule that can help make agriculture healthier and more sustainable.

## 2. Biosynthetic Pathway of Melatonin

Tryptophan, an amino acid that is synthesized from chorismic acid in plants, is the origin of the melatonin biosynthesis pathway [4,72,73] in animal and plant cells, but differs in some steps. In animal cells, tryptophan is converted to 5-hydroxytryptophan by tryptophan hydroxylase (TPH), an enzyme that apparently has not been identified in plants [74,75].

In plants, tryptophan is converted to tryptamine by the enzyme tryptophan decarboxylase (TDC) (Figure 1). Tryptamine is then converted into 5-hydroxytryptamine (serotonin) by tryptamine 5-hydroxylase (T5H), an enzyme that has been extensively studied in rice, and which could act on many substrates. Serotonin was *N*-acetylated by serotonin *N*-acetyltransferase (SNAT). *N*-acetylserotonin is then methylated by acetylserotonin methyl transferase (ASMT), a hydroxyindole-*O*-methyltransferase that generates melatonin. In plants, methylation of *N*-acetylserotonin can also be performed by a caffeic acid *O*-methyltransferase (COMT), a class of enzyme that can act on a variety of substrates, including caffeic acid and quercetin [76]. In plants, serotonin may also be transformed into 5-methoxytryptamine by ASMT/COMT to generate melatonin after the action of SNAT [77]. This route would occur in senescence and/or stress situations [75,78]. Figure 1 shows the steps of the biosynthesis of melatonin in animals (mainly mammals) and plants.

## 3. Physiological Roles of Phytomelatonin

The functions of phytomelatonin in plants have been widely studied, observing that phytomelatonin is a protective biomolecule that activates tolerance and resistance responses in plants, playing an important role against biotic (bacteria, viruses, fungi, insects, parasitic nematodes and weeds) and abiotic stressors (salinity, drought, waterlogging, UV-radiation, heavy metal, heat, cold, mineral deficit/excess, and pesticides) [45,47,58].

Melatonin/phytomelatonin plays an important role in plants as a large antioxidant molecule, similar to how it occurs in animal cells, generally acting as an excellent free radical scavenger, specifically on ROS and RNS [26,28,79]. The direct antioxidant activity of melatonin that neutralizes several ROS/RNS and other free radical species that can damage cells was demonstrated in several in vitro and in vivo models [80,81,82,83]. It also activates the antioxidant redox response, upregulating various transcription factors that trigger the expression of antioxidant enzymes such as superoxide dismutases, catalases, peroxidases, and those involved in the ascorbate-glutathione cycle, among others [28,73,84].

Phytomelatonin is considered a plant master regulator because it regulates the levels and actions of plant hormones. The endogenous levels of auxin, gibberellins (GA), cytokinins (CK), abscisic acid (ABA), ethylene (ET) and other phytohormones such as brassinosteroids, jasmonates (JA) and salicylates (SA) are affected by the action of phytomelatonin through up- or downregulation of transcripts of some biosynthesis/catabolism enzymes, and also hormone-related regulatory factors [47,57,85].

On the other hand, it should be noted that photosynthesis, photorespiration, and stomatal regulation, which are key pieces of the water and carbon economy in plants, are strongly regulated by phytomelatonin. Moreover, the metabolic pathways of carbohydrates, lipids and nitrogen and sulfur compounds are modulated by phytomelatonin, including the osmoregulatory response in stressful situations. In secondary metabolism, phytomelatonin induces the biosynthesis of flavonoids, anthocyanins and carotenoids, among other molecules [44,51,62].

Phytomelatonin also promotes the rooting process for primary and adventitious roots [86], and it regulates leaf senescence, delaying it [87]. In fruit post-harvest, phytomelatonin regulates ethylene and lycopene content, as well as cell wall-related enzymes and primary and secondary metabolism. It also helps preserve cut flowers, delaying senescence, and induces parthenocarpy in fruiting [48].

Due to the ability of phytomelatonin to protect plants against stresses, this molecule could be used as a safener in crops. This agronomical application consists of using phytomelatonin in combination with pesticides, not only to increase the plant tolerance to the possible stress caused by the pesticide (abiotic stress) but also to increase the pesticide efficacy against the biotic stressor [53,88,89,90]. Another application of the effect of phytomelatonin under abiotic stresses is its use postharvest. Phytomelatonin reduces the effect of cold storage of vegetables and fruits by minimizing the damage caused by ROS, improving the quality and commercial shelf-life of fruits and vegetables [91,92,93,94].

Lastly, its role in bacterial, fungal, and viral pathogenic infections should be emphasized. Phytomelatonin slows damage and stimulates systemic acquired resistance (SAR), which contributes to increasing both crop health and postharvest health quality [48,52,61,93,95].

## 4. Melatonin in Plant Disease Biocontrol

There are different approaches that can be used to prevent, mitigate, or control plant diseases. One of them is the application of chemical pesticides, which are used both for preventive as well as for curative disease management [96,97]. However, there is a concern about the negative effects of chemical pesticides due their possible harmful effects on human health, the environment, as well as their effect on the promotion of new resistant pathogens [98,99,100].

There is increasingly stricter legislation in relation to the accessibility and use of efficient pesticides and, therefore, their use is currently declining [101]. Consequently, one study has focused its efforts on developing alternatives to synthetic chemicals for pests and diseases control, some of these being alternatives to so-called biological control.

The term biocontrol refers to the use of naturally occurring (micro-)organisms to control plant diseases or pests [102]. The organism that suppresses the pest or pathogen is found widely in nature, including bacteria, fungi, viruses, yeasts, and protozoans. It can control plant diseases directly, which can be achieved through parasitism, antibiosis, or competition for nutrients or infection sites, or indirectly, where the biocontrol organism induces plant-mediated responses allowing the plant to react faster and more efficiently in case of subsequent pathogen attack [103].

The induced resistance strategy in plants can be indirectly carried out, not only through an organism but also using elicitors; that is, a natural molecule that mimics a pathogen attack or a state of danger, also by living organisms. Plant-induced resistance may represent an interesting strategy for crops [104].

Considering this, melatonin can be an excellent candidate to be used in biocontrol treatments as elicitor molecule. There are currently several studies conducted on the effect of melatonin treatment in the control or reduction of infectious diseases in plants, such as those caused by fungi, bacteria, and viruses. Many of these biocontrol studies are collected in Table 1.

Plant viruses produce local lesions and also cause systemic damage, resulting in malformations, stunting, and chlorosis in plant tissues, even if their hosts are often biotrophic pathogens [68,95]. Viral diseases in plants cause severe economic losses due to agricultural production and have hindered sustainable agricultural development globally for a long time. Unlike diseases induced by fungi and bacteria, viral infections are more difficult to control once the plants are infected [144,145].

In recent decades, various strategies have been developed to control viral diseases, including the breeding of virus-resistant/tolerant cultivars by conventional breeding techniques [146] and the use of eco-friendly chemical compounds that induce systemic resistance [143].

Chemical priming may be considered a timely and successful management technique to induce resistance/tolerance to viruses of plants. Several eco-friendly compounds that are considered non-toxic, biodegradable, and also biocompatible oligomers, such as proteins, polysaccharides and small molecules (alkaloids, flavonoids, phenolics, essential oils) from plants, proteins and polysaccharides from microorganisms, polysaccharides from algae and oligochitosan from animals, can be used to induce plant resistance to viruses [147,148].

Concerning virus infection in plants, there are a few reports (since the first studies were conducted in 2019) based on the study of the interaction of plant viruses with melatonin and its possible role as an inducer of viral resistance in plants. One of these studies was conducted by Chen and cols. (2019) on the potential role of melatonin in the eradication of *Apple Stem Grooving Virus* (ASGV) infection in in vitro Gala apple cultivars [141]. Apple is generally propagated vegetatively by grafting, resulting in transmission of the virus from generation to generation; therefore, it is important to obtain virus-free apple planting materials. Apple shoot segments excised from ASVG virus-infected shoots were cultured in various media supplemented with 0, 10, 15, or 20 µM melatonin and were maintained at 22 °C under a 16 h photoperiod. Ten samples were included in each treatment of three replicates. Treatments of 15 µM melatonin were the most efficient in promoting the number and length of the shoots, as well as the high level of endogenous hormone indole-3-acetic acid. On the other hand, in vitro culture of the virus-infected shoots tips in the medium with 15 µM melatonin resulted in 95% of these shoots being virus-free, while no virus-free shoots were obtained in shoot tips of the virus infected shoots cultured without melatonin. In addition, those plants that continued to be infected, even with 15 µM melatonin in the medium, showed a lower viral load than infected plants grown without melatonin. The virus localization showed that exogenous application of melatonin enlarged the virus-free area of the virus-infected shoot tips. The authors concluded that the exogenous application of melatonin can efficiently eradicate ASGV, being the frequency of the virus eradication related to the melatonin concentration used and the culture time duration on melatonin-containing shoot proliferation medium. Inclusion of 15 µM melatonin in the medium to proliferate shoots for 4 weeks followed by shoot tip culture was found to efficiently eradicate ASGV. This procedure produced 100% of survival and 85% of shoot regrowth levels, and also 95% of virus-free plants in shoot tip culture. The application of melatonin treatments may provide an alternative means for the eradication of plant viruses and could even be used to produce virus-free plants as an interesting biotechnological approach.

Resistance to *Rice Stripe Virus* (RSV) in rice plants treated with melatonin has been studied [142]. The optimum concentration of melatonin and SNP (sodium nitroprusside used as a nitric oxide (NO)-releasing reagent) to reduce disease incidence in the RSV-suscetible *Nipponbare* rice cultivar has been screened. The soil of the 14-day-old rice seedling pots was kept as dry as possible, followed by the application of the different treatments, 0.1, 1, 10, and 100 μM of melatonin or 10, 50, 100, 500 and 1000 µM of SNP. Rice seedlings are placed in the dark for 12 h. After that, they were inoculated with RSV for 3 days. The plants were then transferred to the soil in the greenhouse. Thirty plants were used for each treatment. The results showed that both (melatonin and SNP) can reduce disease incidence in a concentration dependent manner, with the largest effect being observed with 10 μM melatonin and 100 μM SNP. Therefore, the application of exogenous melatonin and NO can promote rice resistance to RSV. Additionally, both compounds positively modulated the expression of two genes (OsPR1b and OsWRKY45 involved in the induction of PR genes (pathogenesis related protein)), indicating that melatonin and NO are able to enhance the plant disease-resistance genes in RSV disease. So, Lu et al. (2019) also quantified the endogenous melatonin levels in two rice cultivars, *Nipponbare* and *Zhendao-88* (susceptible and resistant cultivars to RSV, respectively) after RSV infection. The data showed that resistance to RSV was improved by increased endogenous melatonin and NO production, and established that melatonin was responsible for rice resistance to RSV infection by inducing NO. The authors conclude that rice resistance to RSV can be improved by increasing melatonin through a NO-dependent pathway. The authors postulate that increased melatonin in the resistant cultivar *Zhendao-88* could lead to more NO, which might lead to more SA, which may be the explanation for the increased resistance of this cultivar to RSV [142].

Zhao et al. (2019) examined the plant resistance to melatonin-mediated *Tobacco Mosaic Virus* (TMV) in local infection of *Nicotiana glutinosa* and systemic infection of *Solanum lycopersicum*. For this purpose, the seeds were sown in pots and grown to produce seedlings in an insect-proof net greenhouse under a 16 h photoperiod. Treatments consisted of applying thirty mL of melatonin solutions at 0, 50, 100, 200 and 400 μM to the roots of seedlings at the 5–6- leaf stage grown. After one or two melatonin applications, the leaves were mechanically inoculated with TMV. Ten seedlings were used for each treatment in three independent experiments. The exogenous application of 100 μM melatonin increased the anti-virus infection activity to 37.4% in virus-infected *N. glutinosa* plants. The same treatment significantly reduced the relative levels of virus RNAs and increased the relative expression levels of PR1 and PR5 genes in virus-infected *S. lycopersicum*. Melatonin treatment induced considerable SA and NO accumulation but did not significantly affect the production of hydrogen peroxide in the virus-infected *S. lycopersicum* plants. Additionally, a *N. tabacum* transgenic plant for SA hydroxylase (nahG) showed that the relative RNA level of TMV and virus titers were not reduced by melatonin treatment, indicating the relevance of SA in the melatonin-mediated response. Therefore, melatonin-mediated virus resistance was depressed when cPTIO (a NO scavenger) was used in the application of melatonin, thus clarifying the positive effect of melatonin treatment in improving virus resistance [71].

In another recent study, the possible antiviral activity of melatonin and SA against *Alfalfa Mosaic Virus* (AMV) on eggplants was evaluated [143]. Briefly, eggplant (*Solanum melongena*) seeds were grown in plastic pots in growth chambers with a photoperiod of 12 h. The temperatures in the light and dark period were 27 °C and 23 °C, respectively. After 14 days of growth, the plants were treated with 100 µM melatonin or SA and the dissolutions were sprayed on leaves until run-off. After three days, plants foliar-sprayed were inoculated with AMV to study the response to infection. Five replications were made of each treatment. Foliar spray treatment of 100 µM melatonin or SA substantially reduced the virus titers and the AMV-disease severity in the symptomatic leaves, resulting in a significant increase in the morphological criteria analyzed such as shoot- and root-length, number of leaves, leaf area, and leaf biomass, and also in biochemical parameters such as chlorophyll and carotenoid content, antioxidant enzyme activities, and gene expression of some enzymes as glutathione reductase (GR), dehydroascorbate reductase (DHAR), monodehydroascorbate reductase (MDHAR), chitinase (PR3) and mitogen-activated protein kinase (MPK), compared to the untreated CMV-infected plants (plants control). On the other hand, treatment with melatonin and SA reduced the oxidative damage caused by AMV, decreasing hydrogen peroxide, superoxide anion, hydroxyl radical, and malondialdehyde levels. The authors proposed that melatonin and SA are eco-friendly compounds that could be used in antiviral treatments.

## 5. Melatonin-Integrated Plant Biocontrol Model

Plants have both passive and active defense mechanisms to face the attack of pathogens. The main passive or constitutive defense barriers that they present are the waxy cuticle of the leaves, the cell wall, and the synthesis of secondary metabolites. These impediments, both physical and chemical, represent the first line of defense against infection. However, there are pathogens capable of defeating this first line of defense. In these cases, plants activate an inducible defensive response, called activated immunity, in order to restrict pathogen proliferation. Plants have developed two levels of pathogen detection: the first level of immunity consists of detecting relatively conserved pathogen molecules called pathogen-associated molecular patterns (PAMPs) through plant protein recognition receptors (PRRs). This resistance response is known as PAMP-triggered immunity (PTI). However, there are pathogens capable of overcoming this first level of response, through the secretion of effector proteins. Then, a second level of immunity is activated that involves recognition by intracellular receptors of these pathogen virulence molecules or effectors (R). This receptors have nucleotide-binding domains and leucine-rich repeats (NLRs), which activates effector-triggered immunity (ETI), which is generally accompanied by responses of hypersensitivity (HR) and, finally, of programmed cell death (PCD) to restrict biotrophic cellular pathogens and viruses [149].

Different plant hormones act downstream of ETI or PTI activation as central players in triggering the plant immune signaling network, playing a vital role in resistance [150,151,152]. This hormonal network including SA, ethylene, and JA is crucial in the resistance response to pathogens [153]. SA signaling positively regulates plant defense against biotrophic pathogens which require living tissue to complete their life cycle, whereas ET/JA pathways are commonly required for resistance to necrotrophic pathogens, which degrade plant tissues during infection, as well as in herbivorous pests [154,155].

In addition, other plant hormones such as auxin and ABA, originally described for their role in regulating plant growth processes and the abiotic stress response, have emerged as crucial players in plant-pathogen interactions [156,157,158]. All phytohormone pathways are linked together in a huge complex network (hormonal network). For example, ET, ABA, auxin, GA, and CK pathways are considered to be hormone modulators of the SA–JA signaling backbone [151].

Regarding melatonin as a possible pharmacologic biocontrol agent, a significant increase in the expression of genes involved in ET, JA, CK, GA and auxin metabolism and signaling has been widely described, and this hormonal cross-talk can lead to a high tolerance response against various abiotic and biotic stresses [95,159]. Melatonin would carry out the activation of the two types of immunity mentioned above, the one mediated by PTI in response to PAMP, and the one generated by response to effectors (ETI), in addition to activating signal molecules such as NO and SA, and also ROS, which would serve to protect the plants from a severe attack [129]. Various studies on fungal infection indicated that melatonin is capable of directly causing pathogen growth inhibition, and also improving the defensive capacity of the host plant, upregulating defense genes, involving ROS scavenging and NO production. On the other hand, it has been described that melatonin causes an increase in the levels of cellulose, xylose and galactose of the cell wall in *A. thaliana* leaves infected by bacteria, and an increase in callous deposition, all of which implies a reinforcement of the physical barrier against the invasion of pathogens [67,71,95,130,160].

In the proposed model (Figure 2), the interaction between pathogens (such as virus) and melatonin-mediated pathogen responses is illustrated. The model includes the participation of PRs in the pathogen response [71], taking the melatonin-ROS-NO triad [85], a key player in the control of the phytohormonal network, as an intermediary in the expression of PR genes, such as PR1 and PR5, to improve plant resistance to a virus. In contrast, there are no studies on the role of melatonin in regulating the expression of the genetic material of pathogenic viruses in plants.

## 6. Conclusions

Few studies have addressed the use of melatonin as a biocontrol agent for plant diseases caused by viruses. From the existing information so far, it is concluded that exogenous melatonin treatments can be used to reduce the incidence of disease, reduce symptoms, and even eradicate virus proliferation.

Melatonin treatments helped plant resistance to *Apple Stem Groove Virus* (ASGV), *Tobacco Mosaic Virus* (TMV), *Alfalfa Mosaic Virus* (AMV) and *Rice Stripe Virus* (RSV), limiting rapid viral spread by inhibiting viral movement and reducing virus titer and plant-viral symptomatology with a minor global affectation.

Treatments with melatonin alone or together with SA or NO caused an increase in morphological parameters (shoot, root length, number of leaves, leaf area, and leaf biomass), chlorophyll and carotenoid content, antioxidant enzymes, and gene expression of some antioxidative enzymes compared to those infected untreated plants. Therefore, the treatment also reduced the oxidative damage caused by viruses by reducing ROS (hydrogen peroxide, superoxide anions, hydroxyl radicals) and by lipid peroxidation (malondialdehyde).

Regarding, the possible mechanism of action, it is observed that the exogenous application of melatonin induces an increase in the levels of the defense hormones (SA, JA and ET) and the signaling molecule NO. The upregulation of pathogenesis related genes such as PR1 and PR5 occurs when infected plants are treated with melatonin and/or NO modulators. Other genes such WRKY-45 also were significantly induced by melatonin and NO. These results suggest that resistance to virus attack can be improved by increasing SA and NO levels through exogenous melatonin.

To conclude, melatonin is a molecule with high potential to be used as an antiviral agent in crops, being non-toxic (eco-friendly molecule) with high possibilities of being used in agricultural and biotechnological practices, making them more sustainable. Further studies in more plant species and viruses must be carried out to confirm these incipient results, and to elucidate more details of the molecular mechanism activated by melatonin in virus-infected plants, as it will allow us to understand the process and determine the most effective applications.

## Figures and Tables

**Figure 1 plants-12-00781-f001:**
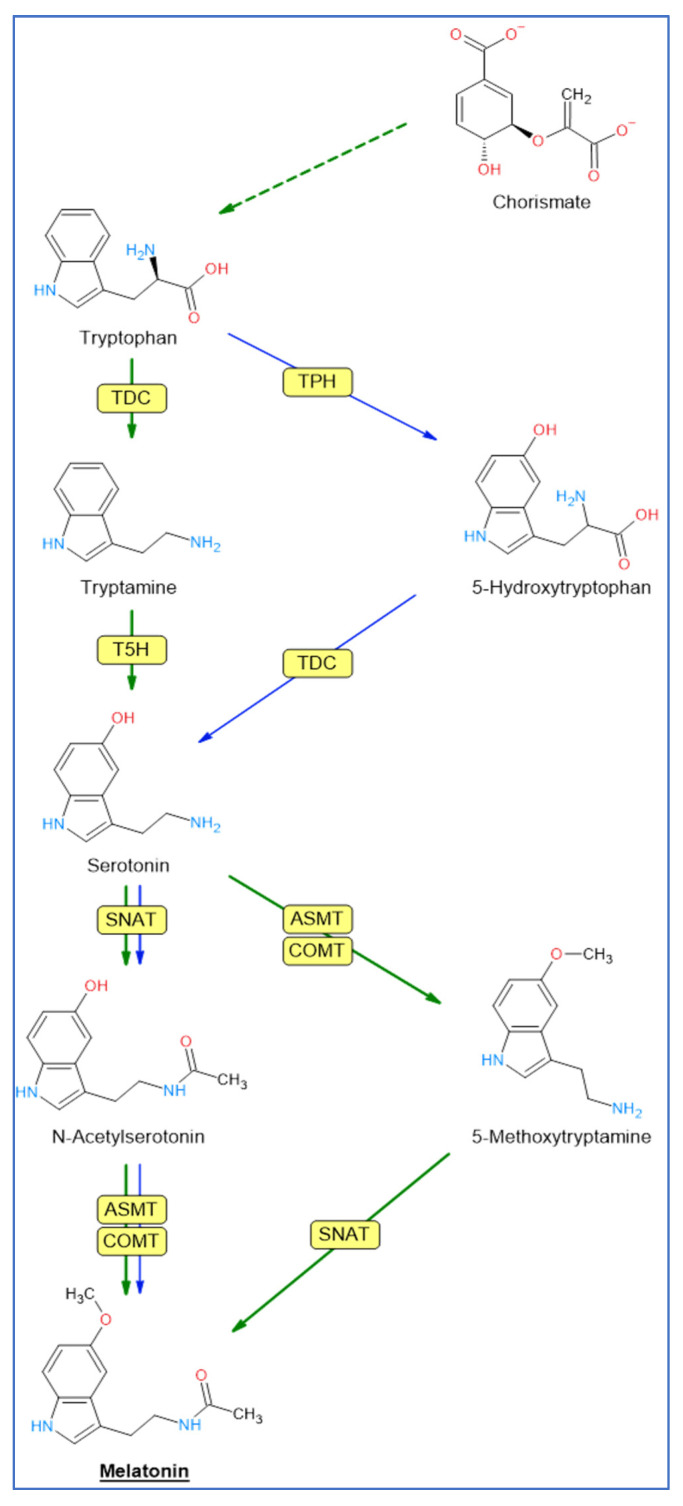
Pathways of melatonin biosynthesis in plants (green arrows) and animals (blue arrows). Enzyme abbreviations: TDC = tryptophan decarboxylase; TPH = tryptophan hydroxylase; T5H = tryptamine 5-hydroxylase; SNAT = serotonin *N*-acetyltransferase; ASMT = acetylserotonin methyl transferase; COMT = caffeic acid *O*-methyltransferase.

**Figure 2 plants-12-00781-f002:**
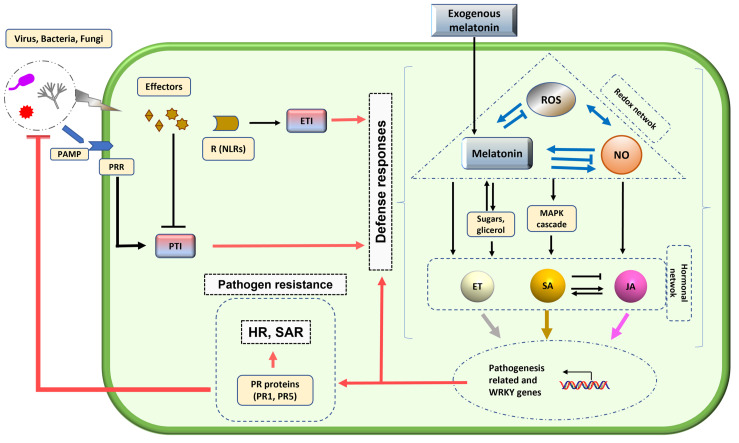
Model proposed for the interaction between plant pathogens (fungi, bacteria and virus) and defense responses mediated by melatonin. When a pathogen attacks plants, a defense system is activated: the pathogen-associated molecular patterns (PAMPs) of pathogens are recognized by the plant sensors called Pattern Recognition Receptors (PRRs) that trigger PTI (PAMP triggered immunity). On the other hand, effectors secreted by pathogens to facilitate infection (block PTI) are recognized by protein receptors (R) that contain nucleotide-binding domains and leucine-rich repeats (NLRs), triggering ETI (Effector-triggered immunity). This recognition activates a cascade of defense responses within the plant cell in which melatonin participates through the production and control of reactive oxygen species (ROS) and reactive nitrogen species (RNS), mainly nitric oxide (NO). At the same time both activate the melatonin endogenous biosynthesis, MAPK kinases cascade, sugars (cell wall reinforcement), defense hormones as ethylene (ET), salicylic acid (SA), jasmonic acid (JA), among others. All this activates the expression of WRKY and defense-related genes as plant resistance genes (PR1, PR5, and others), and consequently systemic acquired resistance (SAR) and the hypersensitive response (HR) are triggered. In this response process, treatment with exogenous melatonin plays an important role, helping to reinforce the plant’s defense, since it acts by increasing the endogenous level of melatonin in the plant cell. As a result, plants reinforced the melatonin-mediated protection against pathogens.

**Table 1 plants-12-00781-t001:** Effects of treatment with exogenous melatonin on plants infected by pathogens (fungi, bacteria, and viruses).

Type of Pathogen	Pathogen Name	Plant Name	Melatonin Dose (µM)	Comments	Ref.
Fungi	*Diplocarpon mali*	*Malus prunifolia*	50, 100, 500	Resistance increased and number of lesions reduced	[105]
Fungi	*Fusarium oxysporum*	*Musa acuminata*	100	Reprogramming of defense-related plant hormones and confers disease resistance	[106]
Fungi	*Phytophthora infestans*	*Solanum tuberosum*	1000, 3000, 6000, 8000, 10,000	Suppression of the virulence and control disease	[107]
Fungi	*Botrytis cinerea* and *Rhizopus stolonifer*	*Fragaria ananassa*	1, 10, 100, 1000	Fruits with lower decay	[108]
Fungi	*Podosphaera xanthii* and *Phythophthora capsici*	*Citrullus lanatus*	1000	Improved plant immunity and suppression pathogen growth	[109]
Fungi	*Phytophthora nicotianae*	*Nicotiana benthamiana*	500, 1000, 2000, 3000, 5000	Suppression of the virulence and control disease	[110]
Fungi	*Verticillium dahliae*	*Gossypium hirsutum*	10, 25, 50, 100	Improved disease resistance	[111]
Fungi	*Botrytis cinerea*	*Solanum lycopersicum*	1, 25, 50, 100	Positive role resistance, regulating JA	[112]
Fungi	*Botrytis cinerea*	*Solanum lycopersicum*	50–500	Improved disease resistance and inhibited gray mold development	[113]
Fungi	*Colletotrichum musae*	*Musa acuminate*	10,000	Delayed senescence and anthracnose incidence	[114]
Fungi	*Penicillium digitatum*	*Citrus reticulata*	50	Decreased disease resistance	[115]
Fungi	*Pseudoperonospora cubensis*	*Cucumis sativus*	100	Reduced index disease	[116]
Fungi	*Fusarium oxysporum*	*Cucumis sativus*	100	Melatonin and arbuscular mycorrhizal enhances resistance	[117]
Fungi	*Colletotrichum gloeosporioides*	*Capsicum annuum* L. and *A. thaliana*	50, 100, 500	Mitigated the infection	[118]
Fungi	*Sclerotinia sclerotiorum*	*Brassica rapa* ssp. Pekinensis	50, 100	Induce defense mechanisms	[119]
Fungi	*Aspergillus flavus*	*Pistacia vera* cv. Akbari	1000	Alleviated oxidative stress and fungal decayed, decreased mycotoxin	[120]
Fungi	*Phytophthora nicotianae*	*Nicotiana benthamiana*	50	Induced immune responses	[121]
Fungi	*Fusarium oxysporum* and *Penicillium brevicompactum*	*Zingiber officinale Roscoe*	100	Reduced postharvest decays	[122]
Fungi	*Peronophythora litchii*	*Litchi chinensis*	250	Improved disease resistance and restricted lesion expansion	[123]
Fungi	*Botrytis elliptica*	*Lilium cultivar* “Sorbonne”	20, 200, 2000, 20,000	Improved plant resistance by MAPK signaling cascade	[124]
Fungi	*Magnaporthe oryzae*	*Oryza sativa* and *Hordeum vulgare*	5000, 10,000	Reduced disease severity and inhibited pathogen growth	[125]
Bacteria	*Pseudomonas syringae* pv. *tomato* DC3000	*Arabidopsis thaliana* and *N. benthamiana*	1, 10	Molecule signaling of defense and inhibition of pathogen propagation	[126]
Bacteria	*Pseudomonas syringae* pv. *tomato* DC3000	*A. thaliana*	1, 2	Increased resistance against pathogen, SA dependent	[127]
Bacteria	*Pseudomonas syringae* pv. *tomato* DC3000	*A. thaliana*	20	Involvement of sugars and glycerol in melatonin-mediated innate immunity	[128]
Bacteria	*Pseudomonas syringae* pv. *tomato* DC3000	*A. thaliana*	20	Improvement of disease resistance by nitric oxide (NO)	[129]
Bacteria	*Pseudomonas syringae* pv. *tomato* DC3000	*A. thaliana*	50	Improved plant resistance by cell-wall reinforcement	[130]
Bacteria	*Pseudomonas syringae* pv. *tomato* DC3000	*A. thaliana* and *N. benthamiana*	1	Improved plant resistance by MAPK signaling cascade	[131]
Bacteria	*Pseudomonas syringae* pv. *tomato* DC3000	*A. thaliana*	1	Improved plant resistance by MAPK signaling cascade	[132]
Bacteria	*Xanthomonas oryzae* pv. *oryzae*	*Oryza sativa* and *N. benthamiana*	861	Reduces pathogenicity and incidence disease	[133]
Bacteria	*Xanthomonas oryzae* pv. *oryzae*	*Oryza sativa*	86	Improved plant resistance by NO and activation of defense-related genes	[134]
Bacteria	*Candidatus liberibacter* and *Diaphorina citri*	*Citrus sinensis*	50–1000	Increased disease resistance by modulation hormonal pathways	[135]
Bacteria	*Bacillus cereus*, *B. licheniformis* and *B. subtilis*	*Solanum lycopersicum* var. *Cerasiforme*	5–10,000	Reduces pathogenicity and incidence disease by ethylene and activation of defense-related genes	[136]
Bacteria	*Pseudomonas syringe* pv. *tomato (Pst)* DC3000	*Panax notoginseng* and *A. thaliana*	10	Reduces pathogenicity and incidence disease	[137]
Bacteria	*Pseudomonas syringe* pv. *tomato (Pst)* DC3000	*A. thaliana*	10, 50	Alteration bacterial resistance in some defense-related hormone signaling	[138]
Bacteria	*Xanthomonas axonopodis* pv. *Manihotis*	*Manihot esculenta Crantz*	50, 100, 150	Increased disease resistance by ethylene	[139]
Bacteria	*Pseudomonas syringae* pv. *Lachrymans*	*Cucumis sativus*	10,100, 1000, 10,000	Reduces pathogenicity and incidence disease	[140]
Virus	*Tobacco Mosaic Virus* (TMV)	*Nicotiana glutinosa* and *S. lycopersicum*	100, 200	Improve plant resistance to infection via SA and NO	[71]
Virus	*Apple Stem Grooving Virus* (ASGV)	*Malus domestica*	10, 15, 20	Efficient eradication plant virus	[141]
Virus	*Rice Stripe Virus* (RSV)	*Oryza sativa*	10	Improve disease resistance	[142]
Virus	*Alfalfa Mosaic Virus* (AMV)	*Solanum melongena*	100	Promotion resistance infection	[143]

## Data Availability

Not applicable.

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
