# Peer review of "Melatonin as a Possible Natural Anti-Viral Compound in Plant Biocontrol"

_plants, 2023, doi:10.3390/plants12040781_

Round 1

Reviewer 1 Report

The manuscript is mostly very well written. I suggest a estensive revision of the paragraph 4 that in my opinion is too long and out of the focus of the paper. In particular:

Paragraph 4

The introduction to the problem (lines 148-183) is too long and out of scope of the paragraph I suggest to reduce it.

Paragraph 4.1

Since the title of the paper was focused on viral effect, I think that that part could be eliminated. I suggest to maintain the table to help interested reader to find more about the use of melatonin. 

Paragraph 4.2

Since the title of the paper was focused on viral effect, I suggest the AAs to add more information especially on how the Melatonin treatments were done, how many plants were treated, etc.…

Author Response

Comments and Suggestions for Authors

The manuscript is mostly very well written. I suggest a estensive revision of the paragraph 4 that in my opinion is too long and out of the focus of the paper. In particular:

Paragraph 4

The introduction to the problem (lines 148-183) is too long and out of scope of the paragraph I suggest to reduce it.

Answer: According to the reviewer's suggestion, the text between lines 148 and 183 has been reduced.

Paragraph 4.1

Since the title of the paper was focused on viral effect, I think that that part could be eliminated. I suggest to maintain the table to help interested reader to find more about the use of melatonin. 

A: According to the reviewer's suggestion, the paragraph 4.1 (from line 192 to 213) has been removed.

Paragraph 4.2

Since the title of the paper was focused on viral effect, I suggest the AAs to add more information especially on how the Melatonin treatments were done, how many plants were treated, etc.…

A: According to the reviewer's suggestion, information on the different melatonin treatments and how they have been applied has been added for each of the four articles on plant-virus studies.

Reviewer 2 Report

This is a comprehensive review regarding the potential application of melatonin to control plant diseases. The review is generally well written and could be useful to provide general information on the effects of melatonin on plant defense response against pathogenic agents and the underlying mechanism of these processes. I recommend the publication of this work. Below are some my comments.

-      As the review also covers the information regarding the effects of melatonin on plant defense against fungus and bacteria, It seems to be good to extend/modify the title, abstract and conclusion to include fungal and bacterial diseases.

-       It is necessary to briefly discuss the economic and technical feasibility of melatonin application in the agricultural fields (large scale application).

-       Lane 343-347. The sentences need to be corrected for clarity.

-       Lane 349. Change “appears” to “is illustrated”

Author Response

Comments and Suggestions for Authors

This is a comprehensive review regarding the potential application of melatonin to control plant diseases. The review is generally well written and could be useful to provide general information on the effects of melatonin on plant defense response against pathogenic agents and the underlying mechanism of these processes. I recommend the publication of this work. Below are some my comments.

Point 1- As the review also covers the information regarding the effects of melatonin on plant defense against fungus and bacteria, It seems to be good to extend/modify the title, abstract and conclusion to include fungal and bacterial diseases.

A: According to reviewer 1's suggestion, the section on fungi and bacteria has been eliminated, maintaining the table, so it would not be necessary to include modifications on fungal and bacterial diseases because the text is now even more exclusively adjusted to viruses.

Point 2-   It is necessary to briefly discuss the economic and technical feasibility of melatonin application in the agricultural fields (large scale application).

A: Melatonin is a molecule that is obtained industrially. Its chemical synthesis is simple and highly economical, making it a compound of considerable purity (≈90%) and cheap, so its use in crop and greenhouse applications would be affordable. Regarding its management, the molecule is amphipathic, so it is easily soluble in aqueous and organic media, not being essential the use of adjuvants in foliar treatments. This paragraph has been included in the new version (L427-432).

-Lane 343-347. The sentences need to be corrected for clarity.

A: The sentence has been rewritten, and we hope it is now more understandable (L369-372).

-Lane 349. Change “appears” to “is illustrated”

A: The expression has been changed in the text.

Reviewer 3 Report

The review by Hernández-Ruiz et al. described the synthesis pathway and biological function of melatonin in plants. Further, the authors thoroughly summarized melatonin's function in plant resistance, especially against plant viruses, and proposed it could be a natural anti-viral compound. 

Minor comments:

Line 167: "and include" should be "including"

lines 196-197: "leaves of Brassica rapa infected by Sclerotinia sclerotiorum improved their defense mechanism when were sprayed with melatonin" could be replaced as "the defense of Brassica rapa against infection by Sclerotinia sclerotiorum was improved by spraying with melatonin on leaves"

lines 230-232: the grammar of this sentence should be checked.

Author Response

Comments and Suggestions for Authors

The review by Hernández-Ruiz et al. described the synthesis pathway and biological function of melatonin in plants. Further, the authors thoroughly summarized melatonin's function in plant resistance, especially against plant viruses, and proposed it could be a natural anti-viral compound. 

Minor comments:

Line 167: "and include" should be "including"

A: The expression has been corrected in the text.

Lines 196-197: "leaves of Brassica rapa infected by Sclerotinia sclerotiorum improved their defense mechanism when were sprayed with melatonin" could be replaced as "the defense of Brassica rapa against infection by Sclerotinia sclerotiorum was improved by spraying with melatonin on leaves"

A: According to reviewer 1's suggestion, paragraph 4.1 (from lines 192 to 213) was completely removed, which includes these lines, so this correction cannot be made.

lines 230-232: the grammar of this sentence should be checked.

A: These lines have been modified when the different reviewers have been answered.

Reviewer 4 Report

This manuscript suggests the application of melatonin as a antiviral agent in biocontrol. This suggestion is based on the effect of melatonin on abiotic and biotic stresses, more studied for bacterial and fungal diseases, since for plant viruses only 4 studies exist to date.

comments and suggestions:
line 153: is a concern or are concerns
line 163: altenativeS
Table I: all concentrations of melatonin in micromolar would make table easier to read

line 242: explain better: if infected shoots incubated with melatonin become virus-free, why does the next sentence say that virus decreased in shoot-tips and enlarged areas are virus -free? For me virus-free means NO virus detectable, eradication, not only reduced!
line 258: For RSV reduction of disease is measured by inoculating melatonin treated plants? How were plants treated?
line 277: anti-virus infection activity? rewrite sentence to make it easier to understand
´line 282: N.tabacum transgenic plant overexpresses gene or gene is mutated??How is melatonin applied in this TMV study?
line 291: "and inducing" does make no sense, can it be replaced with "resulting in"?

While talking about plant biocontrol: how would mekatonin be applied to the plants, spray, irrigation,...? this should be added and discussed.

Author Response

This manuscript suggests the application of melatonin as a antiviral agent in biocontrol. This suggestion is based on the effect of melatonin on abiotic and biotic stresses, more studied for bacterial and fungal diseases, since for plant viruses only 4 studies exist to date.

COMMENTS AND SUGGESTIONS:

Line 153: is a concern or are concerns

A: The expression has been corrected as: “is a concern”

line 163: altenativeS

A: The word has been corrected and has been written in the plural.

Table I: All concentrations of melatonin in micromolar would make table easier to read

A: According to the reviewer's suggestion, all melatonin concentrations have been unified in the micromolar unit.

line 242: explain better: if infected shoots incubated with melatonin become virus-free, why does the next sentence say that virus decreased in shoot-tips and enlarged areas are virus -free? For me virus-free means NO virus detectable, eradication, not only reduced!

A: The sentence has been rewritten in the hope that it will be more understandable.

line 258: For RSV reduction of disease is measured by inoculating melatonin treated plants? How were plants treated

A: The article doesn´t explain deeply how apply the treatments to plants. We have tried to gather the data provided by the authors, summarizing more methodologic information.

Line 277: anti-virus infection activity? rewrite sentence to make it easier to understand

A: The authors analyze the melatonin-mediated resistance to local infection by TMV, in N. glutinosa, calculating the anti-TMV activity based on a formula (Zhao et al., 2015) that calculates the inhibition rate (%), which that allows to determine the % of local lesion comparing the control seedlings and those treated with melatonin. According to that, the sentence can be rewrite.

Line 282: N. tabacum transgenic plant overexpresses gene or gene is mutated?? How is melatonin applied in this TMV study?

A: N. tabacum transgenic plant overexpresses gene. Tobacco plants have been transformed with the bacterial gene nahG, which encodes a bacterial salicylate hydroxylase that degrades SA to catechol, so they will show greater susceptibility to pathogens.

As other reviewer suggests, the different melatonin treatments applied has been added for each one of the four virus-related articles.

Line 291: "and inducing" does make no sense, can it be replaced with "resulting in"?

A: The text has been changed as suggested.

While talking about plant biocontrol: how would melatonin be applied to the plants, spray, irrigation, ...? this should be added and discussed.

A: As other reviewer suggests, the different melatonin treatments applied has been added for each one of the four virus-related articles.